# SCALING LAWS FOR TASK-OPTIMIZED MODELS OF THE PRIMATE VISUAL VENTRAL STREAM

## ABSTRACT

When trained on large-scale object classification datasets, certain artificial neural network models begin to approximate core object recognition (COR) behaviors and neural response patterns in the primate visual ventral stream (VVS). While recent machine learning advances suggest that scaling model size, dataset size, and compute resources improve task performance, the impact of scaling on brain alignment remains unclear. In this study, we explore scaling laws for modeling the primate VVS by systematically evaluating over 600 models trained under controlled conditions on benchmarks spanning V1, V2, V4, IT and COR behaviors. We observe that while behavioral alignment continues to scale with larger models, neural alignment saturates. This observation remains true across model architectures and training datasets, even though models with stronger inductive bias and datasets with higher-quality images are more compute-efficient. Increased scaling is especially beneficial for higher-level visual areas, where small models trained on few samples exhibit only poor alignment. Finally, we develop a scaling recipe, indicating that a greater proportion of compute should be allocated to data samples over model size. Our results suggest that while scaling alone might suffice for alignment with human core object recognition behavior, it will not yield improved models of the brain's visual ventral stream with current architectures and datasets, highlighting the need for novel strategies in building brain-like models. [1]

## 1 INTRODUCTION

The advent of neural networks has revolutionized our understanding and modeling of complex neural processes. A particularly active area of study is the ventral visual stream in primates, a key pathway in the brain responsible for processing visual information Goodale & Milner (1992); Grill-Spector et al. (2001); Malach et al. (2002); Kriegeskorte et al. (2008). Neural networks, when trained on extensive datasets, have emerged as the most accurate quantitative tools for simulating the response patterns of neurons within this stream Yamins et al. (2014); Schrimpf et al. (2018). These advanced models offer a precise computational account of how neural mechanisms in the brain give rise to visual perception.

Recent developments in machine learning have emphasized the significance of both the volume of training data and the complexity of model architectures Kaplan et al. (2020); Hoffmann et al. (2022); Zhai et al. (2022); Bahri et al. (2022); Antonello et al. (2023); Muennighoff et al. (2023); Aghajanyan et al. (2023); Isik et al. (2024). These findings raise the question: Can we build better models of the brain by scaling up model architectures and dataset sizes? Recent studies have found that in pre-trained models, the number of parameters and dataset samples respectively seem to improve predictions of fMRI and behavioral measurements (Antonello et al., 2023; Muttenthaler et al., 2023). With the numerous differences between pre-trained models however, the relative contributions of model parameters and dataset size to brain and behavioral alignment are not clear.

In this paper, we examine how scaling – of model parameters and training dataset size – impacts the alignment of artificial neural networks with the primate ventral visual stream. We systematically train models from a variety of architectural families on image classification datasets which allows us to independently control and observe the effects of model complexity and data volume. To capture

---

[1]We open-source all code, as well as checkpoints for our model zoo.

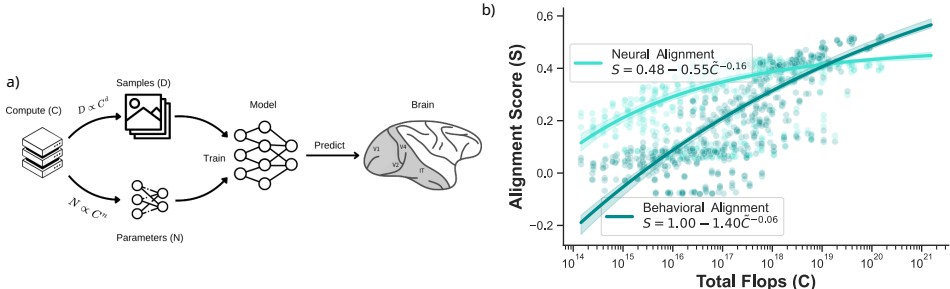

Figure 1: **a)** For a given compute budget ($C$), we determine the scaling laws for maximal neural and behavioral alignment to the primate visual ventral stream. **b)** We find consistent scaling laws for brain and behavioral alignment across over 600 models. While we predict models to approach perfect behavioral alignment of 1 at large scales, the effect of scaling on brain alignment is already saturating at 0.48.

the observed trends, we introduce parametric power-law trends that describe the impact of scale on alignment with behavior and brain regions along the visual ventral stream. We summarize the contributions of this work as follows:

- While scale initially improves alignment, brain alignment saturates whereas behavioral alignment continues to improve.
- Increasing both parameter count and training dataset size improves alignment, with data providing more gains over model scaling.
- Architectures with stronger inductive bias (such as convolutions) and datasets with higher-quality images are more sample- and compute-efficient.
- Fitting parametric power-law curves, we find that model alignment with higher-level brain regions and especially behavior benefits the most from scaling.
- We publicly release our training code, evaluation pipeline, and over 600 checkpoints for models trained in a controlled manner to enable future research.

## 2 RELATED WORK

**Primate Visual Ventral Stream.** The ventral visual stream, a critical pathway in the primate brain, including humans, plays a key role in visual perception, extending from the occipital to the temporal lobes and serving as the "what pathway" for object recognition and form representation Goodale & Milner (1992); Grill-Spector et al. (2001); Malach et al. (2002); Kriegeskorte et al. (2008). Beginning in the primary visual cortex (V1), where basic visual information from retinal ganglion cells is processed, the ventral stream proceeds through areas such as V2, V3, V4, and the inferotemporal cortex (IT), each responsible for increasingly complex features of visual perception Kandel et al. (2000). Despite decades of research and a wealth of brain data, the precise neural mechanisms underlying visual perception are not well understood.

**Modeling the Primate Visual Ventral Stream.** Particular artificial neural networks (ANNs) are the most accurate models of brain responses in the visual ventral stream and associated core object recognition behaviors Schrimpf et al. (2018; 2020). Models optimized for ecologically viable tasks (Yamins & DiCarlo, 2016) in particular have demonstrated strong brain and behavioral alignment (Yamins et al., 2014; Khaligh-Razavi & Kriegeskorte, 2014; Cadena et al., 2019; Schrimpf et al., 2018; Nayebi et al., 2018; Kietzmann et al., 2019; Rajalingham et al., 2018; Zhuang et al., 2021; Geiger et al., 2022) – notably these models are trained purely on image classification datasets, without fitting to brain data.

**Scaling Laws.** Recent advancements in artificial intelligence are driven by scaling the model size and training data. Empirical evidence suggests a power-law relationship between model performance and both model parameters and dataset size, indicating that continued scaling will further

improve performance Kaplan et al. (2020); Cherti et al. (2023); Zhai et al. (2022); Hoffmann et al. (2022); Dehghani et al. (2023); Henighan et al. (2020); Brown et al. (2020); Bahri et al. (2022); Hestness et al. (2017). The power-law exponents enable the optimal allocation of compute between model parameters and dataset samples, such that performance is maximized Kaplan et al. (2020); Hoffmann et al. (2022).

While scaling laws for machine learning *performance* has been extensively studied, the scaling laws for *brain alignment* remain unclear. Recent studies indeed suggest an involvement of both model size and data volume in the functional alignment with brain data Azabou et al. (2023); Benchetrit et al. (2023); Caro et al. (2024); Antonello et al. (2023). Conversely, Muttenthaler et al. Muttenthaler et al. (2023) indicate that sample size is critical for behavioral alignment. We here unify these results, in the realm of the primate visual ventral stream, into quantitative scaling laws for how model and dataset sizes relate to alignment with the brain and behavior.

## 3 METHODS

**Neural & Behavioral Alignment.** To evaluate the alignment of our model with brain function, we utilize a range of benchmarks from Brain-Score Schrimpf et al. (2018; 2020). These benchmarks assess model performance by comparing model activations or behavior with primate neural data using the same images. Specifically, the V1 and V2 benchmarks compare model outputs to primate single-unit recordings from Freeman et al. (2013), using 315 texture images and data from 102 V1 and 103 V2 neurons. For the V4 and IT benchmarks, 2,560 images are used to match model activations to primate Utah array recordings from Majaj et al. (2015), based on data from 88 V4 and 168 IT electrodes. A linear regression is trained on 90% of the images to correlate model and neural data, with prediction accuracy for the remaining 10% evaluated using Pearson correlation, repeated ten times for cross-validation. The behavioral benchmark assesses model predictions for 240 images against primate behavioral data from Rajalingham et al. (2018) using a logistic classifier trained on 2,160 labeled images. Pearson correlation is used to measure the similarity in confusion patterns between model predictions and primate responses. All benchmark scores are normalized to their respective maximum possible values.

We define the model's alignment score $S$ (and an inverse *Misalignment Score* $L = 1 - S$) as the average across the V1, V2, V4, IT, and behavioral benchmark scores. Layers are committed to brain regions based on models trained on a full dataset, and applied to all variants trained with subsampled datasets. As we reused the same neural and behavioral data both to select the optimal model layer for readout and to assess the model's alignment, we validated the benchmark results on a private split of each dataset on Brain-Score. We observed an almost perfect correlation between the results on the private and public splits (Appendix C).

**Scaling Models and Data.** We trained an array of standard models from several architecture families. Specifically, we used ResNet$18, 34, 50, 101, 152$ from He et al. (2016); EfficientNet-B$0, 1, 2$ from Tan & Le (2019); Vision Transformer ViT$T, S, B, L$ from Dosovitskiy et al. (2021); ConvNeXt$T, S, B, L$ from Liu et al. (2022b); CORnet-S from Kubilius et al. (2019); and AlexNet from Krizhevsky et al. (2012). We also trained 33 modified versions of ResNet18: 22 models obtained by scaling the network width from $1/16$ to $4$ times the original size, and 11 models derived by adjusting the depth. Similarly, we trained four additional ConvNeXt and ViT models by scaling the width of the ConvNeXt-T and ViT-S architectures.

For our experiments, we selected two image classification datasets: ImageNet Deng et al. (2009) and EcoSet Mehrer et al. (2021). ImageNet, with millions of labeled images across 1,000 categories, has long been a benchmark in computer vision, designed to challenge and evaluate automated visual object recognition systems. On the other hand, EcoSet is a more recent dataset, designed to provide an ecologically valid representation of human-relevant objects. It contains over 1.5 million images spanning 565 basic-level categories, curated to better reflect the natural distribution of objects in the real world, aligning with human perceptual and cognitive experiences.

To create subsets of ImageNet and EcoSet, we sampled $d \in 1, 3, 10, 30, 100, 300$ images per category. For $d \in 1, 10, 100$, we repeated the runs with three random seeds to ensure robustness. For ConvNeXts (Liu et al., 2022b) and ViTs (Touvron et al., 2022), we used the training recipes developed by the original model authors. The remaining models were trained for 100 epochs using a

minibatch size of 512. We employed a stochastic gradient descent (SGD) optimizer with a cosine decaying learning rate schedule, starting with a peak learning rate of 0.1 and incorporating a linear warm-up phase spanning five epochs. We maintained the momentum at 0.9 and applied a weight decay of $10^{-4}$. Cross-entropy loss was used as the minimization objective. We utilized standard ImageNet data augmentations, specifically random resized cropping and horizontal flipping.

**Scaling Power-Law Curves.** Following previous work on scaling laws Zhai et al. (2022); Hoffmann et al. (2022); Besiroglu et al. (2024), we fit power law functions in the form

$$L = E + AX^{-\alpha} \tag{1}$$

on the data where $L$ is the misalignment score, and $X$ is an independent variable, such as the number of samples seen ($D$), number of parameters ($N$), and the total training floating point operations (FLOPs) ($C$). Coefficients $E$, $A$, and $\alpha$ are found by minimizing

$$\min_{a,e,\alpha} \sum_{i \in [\#\text{Runs}]} \text{Huber}_\delta \left( \text{LSE}(a - \alpha \log X_i, e) - \log L_i \right) \tag{2}$$

where $E = \exp(e)$, $A = \exp(a)$ and LSE is the log-sum-exp operator. We solve Eq. 1 using BFGS minimizer with $\delta = 1e - 3$, and use a grid of initialiations as follows: $e \in \{-1, -0.5, \ldots, 1\}$, $a \in \{0, 5, \ldots, 25\}$, $\alpha \in \{0, 0.5, \ldots, 2\}$.

To capture the slow initial increase in benchmark scores of modern architectures like ConvNeXt and ViT models in the low-data regime, we introduce an additional parameter $\lambda$ to Eq. 2. This parameter allows the fitted curve to saturate at lower scales, better reflecting the observed performance of these models under limited data conditions:

$$L = E + A \left( X + 10^\lambda \right)^{-\alpha} \tag{3}$$

We minimize the modified equation as before, using $\lambda \in 0, 0.5, 1.0, 1.5, 2.0$. To fit the curve described by Eq. 3, we utilize all data points from the ConvNeXt and ViT models. For fitting the remaining curves, we select ConvNeXt and ViT runs that were trained on datasets with either 300 samples per class or the full dataset. This approach ensures that the fitted curves accurately represent the scaling behavior of these architectures across different data regimes.

Furthermore, we would like to describe the misalignment ($L$) as a function of both the model and data size ($N, D$) and predict optimal allocations $N^*$ and $D^*$ by solving

$$(N^*, \ D^*) = \underset{N, \ D}{\arg\min} L(N, \ D) \quad \text{s.t} \quad \text{FLOPs}(N, \ D) = C \tag{4}$$

In that regard, following Hoffmann et al. (2022); Besiroglu et al. (2024) we fit a parametric function of the form

$$\hat{L}(N, D) = E + \frac{A}{N^\alpha} + \frac{B}{D^\beta} \tag{5}$$

where the loss ($\hat{L}$) is a function of parameter count ($N$) and number of samples seen ($D$). In Eq. 5, the first term represents the loss in an ideal data generation scenario (entropy), the second and the third terms reflect the under-performance of a model due to limitations in parameter and data size Hoffmann et al. (2022); Muennighoff et al. (2023). Following the example of Hoffmann et al. (2022), we learn variables $\{E, \ A, \ \alpha, \ B, \ \beta\}$ that characterizes misalignment by solving

$$\underset{e, \ a, \ \alpha, \ b, \ \beta}{\arg\min} \sum_{i \in [\#\text{Runs}]} \text{Huber}_\delta \left( \text{LSE}(a - \alpha \log N_i, b - \beta \log D_i, e) - \log L_i \right) \tag{6}$$

with $\delta = 10^{-3}$ and $E = \exp(e)$, $A = \exp(a)$ $B = \exp(b)$. Initialiations of $b$ and $\beta$ follow $a$ and $\alpha$, respectively.

Both Kaplan et al. (2020); Hoffmann et al. (2022) assume that compute follows the relationship $C(N, D) \approx 6ND$ to predict the optimal allocation of compute ($C$) to $N$ and $D$ using a set of equations with the learned variables mentioned above:

$$N^*(C) = G(C/6)^a, \qquad D^*(C) = G^{-1}(C/6)^b$$

$$\text{where} \qquad a' = \frac{\beta}{\alpha + \beta}, \quad b' = \frac{\alpha}{\alpha + \beta}, \quad G = \left( \frac{\alpha A}{\beta B} \right)^{\frac{1}{\alpha + \beta}} \tag{7}$$

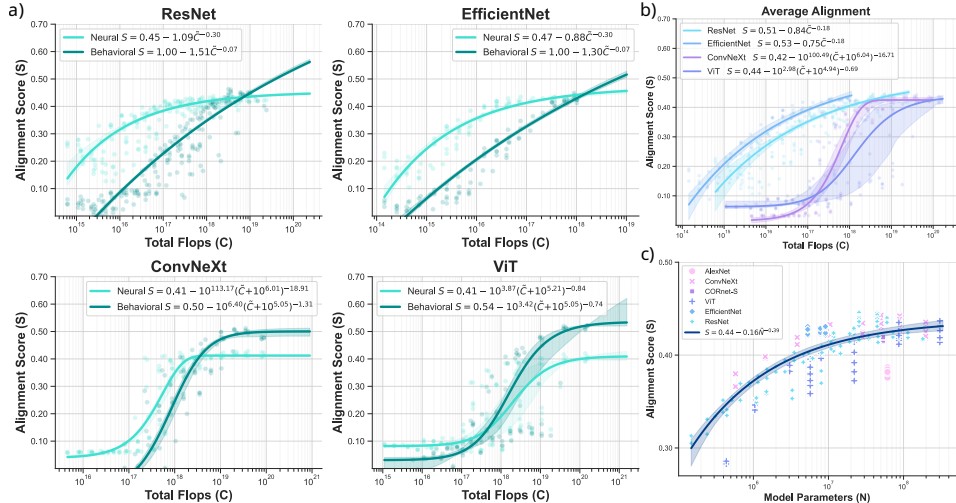

Figure 2: **Scaling Model Size. a)** Neural and behavioral alignments of four architecture families. Models with inductive biases (ResNet, EfficientNet) are more compute-efficient than less constrained models (ConvNeXt, ViT). **b)** Average alignment per model architecture. All models converge to similar alignments. **c)** Increasing parameters improves alignment (models trained on full datasets), but the effects saturate.

However, we observe that $C(N, D) \approx 6ND$ does not hold with different architectures, and various CNN families have a slightly different relationship of $C$, $N$, and $D$. As such, we assume a power-law relationship of the form

$$C(N, \ D) = m(ND)^n \tag{8}$$

where we fit $m$ and $n$ via linear regression of $C$ and $ND$ in log-log scale. Then, the updated equations governing the optimal allocation becomes

$$N^*(C) = G(C/m)^{a'/n}, \qquad D^*(C) = G^{-1}(C/m)^{b'/n} \tag{9}$$

where $a'$, $b'$, and $G$ are calculated as before.

To evaluate the uncertainty of our model fits, we performed bootstrapping with 1,000 resamples. We compute 95% confidence intervals for each point along the fitted curves based on the variability observed across the bootstrapped estimates.

Finally, to avoid large constants during curve fitting, we rescale the variables $C$, $N$, and $D$ by setting $\tilde{C} = C/10^{13}$, $\tilde{N} = N/10^5$, and $\tilde{D} = D/10^4$.

## 4 RESULTS

### 4.1 SCALING DRIVES BEHAVIORAL ALIGNMENT, BUT SATURATES FOR NEURAL ALIGNMENT

Our experiments show a clear and consistent improvement in behavioral alignment as both model size and training dataset size increase. Fig 1.b illustrates this trend across different architectures and scaling axes. The curve $S = 1 - 1.4\tilde{C}^{-0.06}$ converges to perfect alignment score of 1 in the limit of $C$.

In contrast to behavioral alignment, neural alignment with specific brain regions demonstrated saturation as training compute scaled up in size. The curve represented by the formula $S = 0.48 - 0.55\tilde{C}^{-0.16}$ represents a saturation at $0.48$. The diminishing returns in neural alignment imply that merely scaling up models and data is insufficient to achieve better alignment with higher-level neural representations.

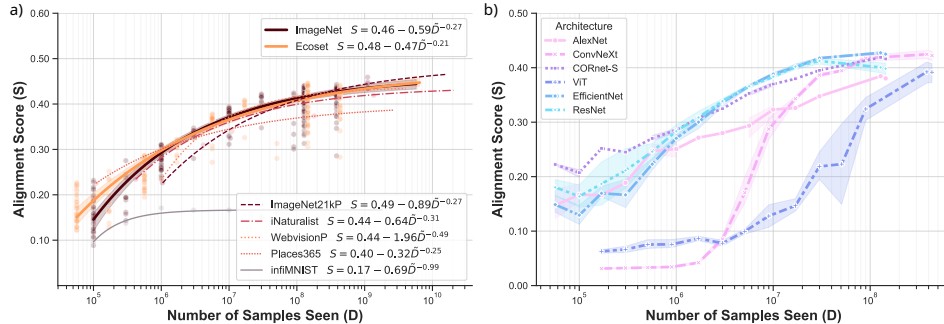

Figure 3: **Scaling Dataset Size. a)** Training on larger datasets enhances brain alignment. The alignment scaling curves derived from ImageNet and EcoSet closely estimate the alignment achieved when using ImageNet21k. In contrast, datasets with specialized image distributions—such as Places365—fall below the alignment scaling laws established by these generalist datasets. **b)** Model families with weaker inductive bias start at a lower alignment and require more data to improve.

## 4.2 ARCHITECTURAL INDUCTIVE BIAS INFLUENCES ALIGNMENT AND SCALING BEHAVIOR

Experimental results indicate that modern architectures, such as ConvNeXt and Vision Transformers (ViTs), exhibit poorer neural alignment compared to models like ResNets and EfficientNets in low data regime. ResNets and EfficientNets, which have stronger inductive biases due to their fully convolutional structures, demonstrate high neural alignment even at initialization. In Fig. 2, alignment score of ResNets and EfficientNets increase steadily with additional compute in the form of training samples, however ConvNeXt and ViT requires more compute in order to start rising.

This difference in initial alignment also affects how the scaling laws evolve for each architecture. Models with weaker inductive biases require more extensive scaling—specifcally in terms of training data—to achieve levels of neural alignment comparable to those with stronger inductive biases. Consequently, the scaling curves for ConvNeXt and ViT models develop differently, highlighting that architectural choices not only impact baseline alignment but also influence the efficiency of scaling strategies.

## 4.3 MORE DATA IS BETTER THAN MORE PARAMETERS

Our analysis reveals that increasing the size of the training dataset has a more significant impact on improving brain alignment than simply enlarging the number of model parameters. While both strategies lead to performance enhancements, the benefits from data scaling exhibit less severe diminishing returns compared to model scaling. Specifically, models trained on larger datasets consistently demonstrate superior neural and behavioral alignment with the primate ventral visual stream, following a predictable power-law relationship.

In contrast, expanding the model size without proportionally increasing the training data results in steeper diminishing returns in alignment performance. Larger models rapidly reach a point where additional parameters do not translate into meaningful improvements. Fig. 2c estimates a saturation level of $0.44$ by scaling model sizes with all samples of training data whereas Fig. 3a predicts maximum alignment of $0.48$ and $0.50$ for ImageNet and Ecoset respectively. This indicates that scaling training datasets overall improves brain alignment better than models scaling. Furthermore, Fig. 4b demonstrates that larger models of the same architecture family require much more samples to achieve the same level of alignment.

To quantitatively capture the joint interaction between data and model scaling, we fitted a parametric curve based on Eq.5, as shown in Fig.4a. This curve effectively models how compute ($C$), dataset size ($D$), and model size ($N$) collectively influence brain alignment. Utilizing the parametric relationships described in Eq. 9, we estimate that additional compute should be allocated following the scaling laws $D \approx C^{0.7}$ and $N \approx C^{0.3}$. These exponents indicate that, for optimal brain alignment, computational resources should be predominantly invested in increasing the dataset size rather than the model size.

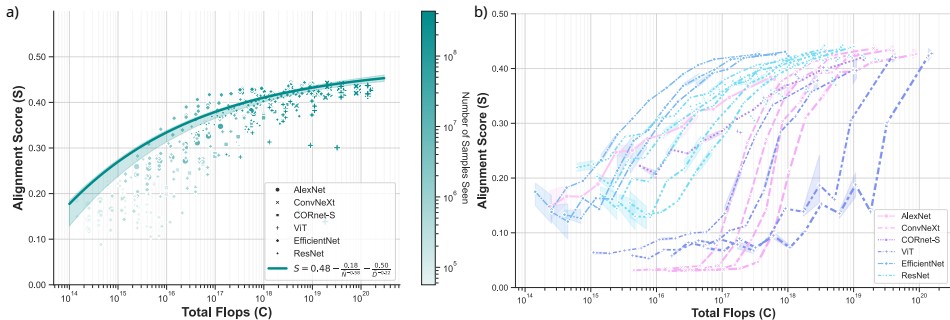

Figure 4: **Optimal Compute Allocation. a)** Alignment as a function of both model and training dataset sizes. Marker size is log-proportional to model size. Compute should be spent 0.3/0.7 on model/dataset size respectively. **b)** Models start out at different alignments but converge to the same saturating point.

### 4.4 ORDERED EFFECT OF SCALE ON ALIGNMENT

Our study reveals a graded effect of scaling on alignment across the cortical hierarchy of the primate visual system. Specifically, we observe that the benefits of increased training compute—achieved through larger datasets and more complex models—vary systematically among different brain regions, reflecting their position in the visual processing pathway. Fig. 5.a illustrates the alignment as a function of training compute across various brain regions. We categorized the models into two groups based on their architectural inductive biases. Group 1 includes most models with strong inductive biases, such as ResNets and EfficientNets. These models start with higher neural alignment scores even at initialization due to their fully convolutional architectures. Group 2 consists of models with weaker inductive biases, specifically ConvNeXt and Vision Transformers (ViTs). These models exhibit lower neural alignment in the low-data regime and require more compute to achieve similar alignment levels.

To quantify the impact of scaling on each brain region, we define the alignment gain per region as $A10^\alpha$ where $A$ and $\alpha$ are parameters of Eq. 2. Our findings indicate that higher regions in the cortical hierarchy show greater benefits from increased compute. Fig. 5b illustrates the alignment gain per region, highlighting how higher cortical areas benefit more from scaling efforts. This ordered effect suggests that regions higher up in the visual hierarchy, such as the Inferior Temporal (IT) cortex and behavioral outputs, gain more substantially from additional data and increased model complexity. In contrast, early visual areas like V1 and V2 exhibit smaller alignment gains with increased compute, indicating a potential saturation effect.

## 5 DISCUSSION

We establish scaling laws governing the effect of model and dataset scale on behavioral and brain alignment with the primate visual ventral stream. While scale is a necessary component for all brain-like models, model architectures with priors such as convolutions, and datasets with high-quality images are more sample efficient, leading to alignment with smaller compute requirements. Scale especially improves alignment with higher-level visual regions, but brain alignment saturates across all conditions tested here whereas behavioral alignment continuously improves with increased scale.

**Platonic representations.** Our results are consistent with the view that representations in deep neural network models are converging. This observation has been likened to Plato's concept of an ideal reality (Huh et al., 2024). Our findings support this view in that neural network models at scale tend to converge toward similar brain and behavioral alignment, regardless of their initial architectural differences. However, we find that the representational convergence in current models is *distinct from representations in the brain*'s visual ventral stream. Intriguingly, behavioral choices on the other hand seem to converge to human-like core object recognition, suggesting that there is no unique solution to human-like behavior. Even models with differing inductive biases, such as ConvNeXt and ViT (which start with poorer neural alignment), eventually converge to the same

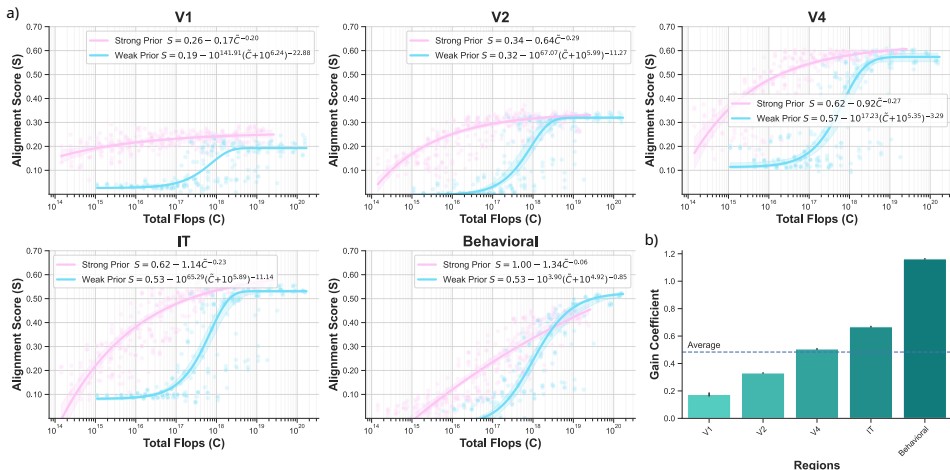

Figure 5: **Graded Effect of Scale across Cortical Hierarchy. a)** Alignment as a function of training compute across different brain regions. Group 1 contains most models except those with low inductive bias (Group 2; ConvNeXt, ViT). **b)** Alignment gain per region, defined as $A10^{\alpha}$. Regions higher in the cortical hierarchy show greater benefits from increased compute (Behavior $>$ IT $>$ V4 $>$ V2 $>$ V1).

.

behavioral, and, saturating brain alignment as models like ResNet and EfficientNet. Building models with representations that are consistent with the brain might thus require substantial changes to current architectures and training approaches.

**Dissociation of behavioral and neural alignment.** Our findings reveal a dissociation between behavioral and neural alignment as models are scaled with more parameters and larger datasets. While behavioral alignment continues to improve consistently with increased model parameters and training data – exhibiting a strong power-law relationship – neural alignment reaches a saturation point beyond which additional scaling yields minimal gains. This divergence suggests that behavioral alignment benefits more substantially from scaling efforts, whereas neural alignment may require alternative approaches beyond merely increasing model size and data volume to achieve further improvements.

This disparity is further highlighted by the correlation between task performance and alignment depicted in Figure 6. Behavioral alignment closely tracks validation accuracy, improving hand-in-hand as models become more accurate. Consistent with prior work(Schrimpf et al., 2018; Linsley et al., 2023), neural alignment eventually saturates, indicating that factors other than task performance influence neural alignment.

**Generalization Beyond Supervised Training.** We assessed whether alternative training paradigms can overcome the limitations observed in neural alignment under supervised learning. Figure 7a illustrates the scaling of alignment as a function of compute spent during self-supervised training of ResNet models using SimCLR Chen et al. (2020) on ImageNet. The results confirm the trends observed in supervised training: behavioral alignment continues to improve with increased compute, following a strong power-law relationship, while neural alignment approaches a saturation point. This consistency suggests that the saturation in neural alignment is not exclusive to supervised learning but may be inherent to the models or datasets employed.

The region-specific breakdown (as illustrated in Figure 13 of the appendix) further reinforces this observation. Even in a self-supervised learning context, higher-level visual areas like IT and behavioral outputs demonstrate more pronounced improvements with increased compute, while early visual areas like V1 and V2 show minimal gains. This suggests that the hierarchical nature of neural alignment is a fundamental characteristic that transcends specific training methodologies.

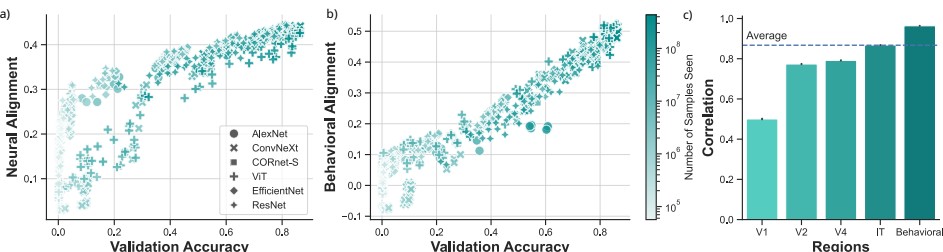

Figure 6: **Correlation between Task Performance and Alignment. a,b)** Correlation between validation accuracy (ImageNet & EcoSet) and brain (a) and behavioral (b) alignment. Behavioral alignment strongly correlates with task performance, whereas neural alignment shows a non-linear trend, reaching saturation. **c)** Pearson's correlation coefficient per region, with all p-values less than $10^{-40}$.

Additionally, we explored the impact of adversarial fine-tuning on alignment performance. In Figure 7b, ResNet models trained on subsets of ImageNet were fine-tuned adversarially for 10 epochs using the Fast Gradient Sign Method (FGSM) (Goodfellow et al., 2015; Wong et al., 2020). Importantly, the scaling curves were estimated solely from the non-adversarial runs, yet the adversarially fine-tuned models exhibited improvements along these existing scaling curves. This indicates that adversarial training can enhance alignment without deviating from the established scaling behavior.

**Impact of Architectural Inductive Biases on Alignment Dynamics.** Our evaluation of alignment during training reveals that the alignment behavior varies significantly across different model architectures. Figure 7.c shows that while various models eventually converge to similar alignment levels with sufficient training, fully convolutional architectures—such as ResNets and Efficient-Nets—exhibit substantially higher alignment scores at the very beginning of training. This early advantage suggests that these architectures possess inherent features that align closely with neural data from the primate ventral visual stream even before learning from data occurs.

Further analysis in Figure 10 of the appendix confirms that this initial high alignment is due to the strong inductive biases present in fully convolutional networks. These biases enable the models to start with representations already well-suited for neural alignment. Figure 11 in the appendix reinforces this finding by demonstrating that models with strong inductive biases achieve higher initial alignment compared to architectures like ConvNeXt and ViT, which have weaker inductive biases.

**Influence of Learning Signals on Alignment Dynamics.** Our investigation reveals that the type of learning signal plays a crucial role in the dynamics of alignment during training. Figure 7d illustrates the alignment trajectories of ResNet50 and ViT-S models trained on ImageNet using supervised learning, SimCLR, and DINO (Caron et al., 2021) methods. Notably, the ViT-S model requires significantly more training steps to achieve the same level of alignment under supervised learning compared to when trained with self-supervised objectives like DINO and SimCLR. In contrast, the ResNet50 model, which possesses strong inductive biases due to its convolutional architecture, exhibits relatively consistent alignment dynamics across different learning signals. This robustness implies that models with strong inductive biases are less affected by the choice of training objective, whereas architectures like ViT-S benefit more substantially from rich, self-supervised feedback to achieve optimal alignment.

**Limitations and Future Directions.** Our study has several limitations. First, the extrapolation of our scaling functions is constrained by the specific range of model sizes and dataset volumes we examined. While we observed power-law relationships between scaling factors and brain alignment, these functions may not generalize beyond the scales tested.

Second, we evaluated a subset of models focusing primarily on standard and modern convolutional neural networks (e.g., ResNets and ConvNeXts), transformer-based architectures (e.g. ViTs) and recurrent networks (CORnet-S). While these architectures cover a range of inductive biases and complexities, they do not encompass the full spectrum of possible neural network designs, such

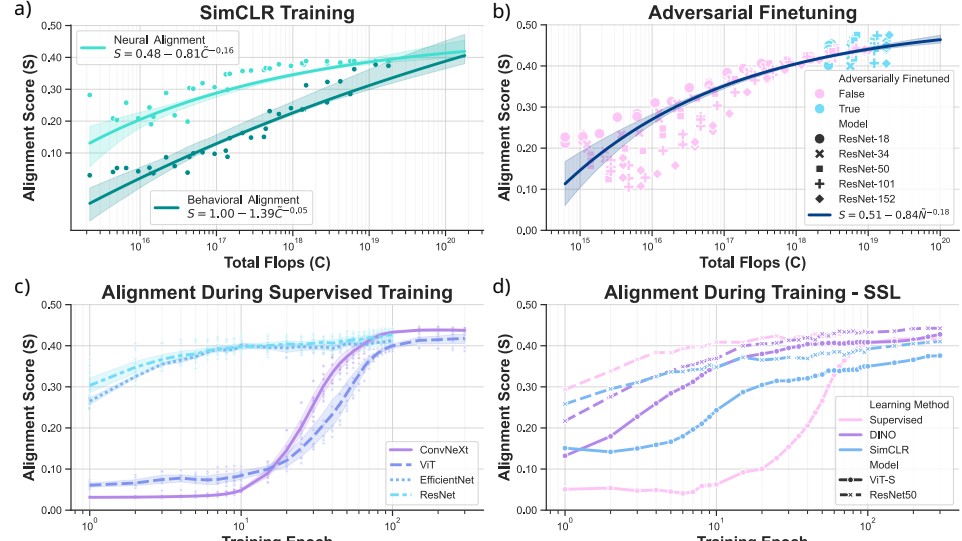

Figure 7: **Alignment Scaling of Alternative Training Strategies. a)** Unsupervised: ResNet models trained with SimCLR improve behavioral alignment with compute, while neural alignment saturates. **b)** Adversarial Robustness: Fine-tuning ResNets with adversarial training (FGSM) enhances alignment along the scaling curve. **c)** Architectural Prior: ResNets and EfficientNets exhibit higher initial alignment in early phases of training, due to strong inductive biases, unlike ConvNeXt and ViT. **d)** Alignment dynamics vary with training objectives, but converge to the same alignment saturation.

as more biologically plausible models. We see scaling laws as an opportunity to extrapolate the alignment of models at scale, even if their current training is compute-constrained.

Third, our experiments utilized a subset of training datasets primarily from ImageNet and EcoSet. Although these datasets are extensive and widely used, they may not capture all the nuances of visual stimuli relevant to the primate ventral visual stream. Therefore, models trained on other datasets might exhibit improved scaling properties.

Taken together, our results demonstrate that while scaling both model parameters and training data size enhances behavioral alignment with human visual perception, it leads to saturation in neural alignment with the primate ventral visual stream. Data scaling proves more effective than model scaling in improving alignment, emphasizing the critical role of extensive and diverse training datasets. We also find that architectural choices significantly influence alignment efficiency, with models possessing strong inductive biases—such as fully convolutional networks—achieving higher neural alignment even at initialization. Additionally, the impact of scaling varies across different brain regions, benefiting higher cortical areas more than early visual areas. These findings suggest that merely increasing scale is insufficient for modeling the intricate neural representations of the brain's visual system. Future work should investigate new approaches, including alternative architectures and training strategies, to develop models that more accurately reflect the complexities of neural processing in the primate visual cortex.

To push neural alignment beyond current saturation levels, future research should explore adversarial training methods that encourage models to learn more robust, brain-like representations. Leveraging biologically inspired architectures such as VOneNets (Dapello et al., 2020) may lead to more compute-efficient models achieving higher neural alignment without extensive scaling. Additionally, investigating co-training with brain data—integrating neural recordings directly into the training process—could enhance both neural and behavioral alignment, paving the way for more accurate and efficient brain-like models.

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

## A   IMPLEMENTATION DETAILS

Our experiments are conducted using the PyTorch framework (Paszke et al., 2019), with Composer (Team, 2021) employed as the GPU orchestration tool to efficiently manage computational resources.

For image augmentations, we leverage the Albumentations Buslaev et al., 2020 library due to its rich set of augmentation techniques, which are crucial for enhancing model robustness and preventing overfitting. In experiments involving self-supervised learning, we use the Lightly Susmelj et al. (2020) library to facilitate the implementation of self-supervised losses, augmentations, and model heads. This library streamlines the process of setting up models for SimCLR and DINO training methods.

To generate adversarial examples for adversarial fine-tuning, we employ the Torchattacks library Kim (2020). Specifically, we use the Fast Gradient Sign Method (FGSM) to create perturbations that challenge the models, aiming to enhance their alignment with neural representations by exposing them to adversarial inputs.

## B   ADDITIONAL IMAGE DATASETS

To further validate our findings across diverse image distributions and to estimate scaling curves across different sample scales, we trained ResNet18 models on subsets of several large-scale image datasets: ImageNet-21k-P, WebVision-P, iNaturalist, and Places365. Below, we provide detailed descriptions of each dataset.

### B.1   IMAGENET21K-P

ImageNet-21k-P is a processed subset of the full ImageNet-21k dataset (Ridnik et al., 2021), which originally contains over 14 million images organized into more than 21,000 categories following the WordNet hierarchy. The "P" denotes a pruned version where classes with insufficient images or noisy labels are filtered out to enhance dataset quality. This results in a refined dataset that maintains the richness of the original ImageNet-21k while improving label accuracy and image relevance. The resulting dataset contains approximately 11 million training images across 10,450 classes.

### B.2   WEBVISION-P

The WebVision dataset (Li et al., 2017) is a large-scale web image dataset designed to provide a real-world, noisy alternative to ImageNet. It originally contains over 16 million images categorized into 5,000 classes. The images are collected from the internet using queries from search engines like Google and Flickr, leading to a dataset that includes label noise, varying image resolutions, and diverse visual contexts. Due to classes with very few available samples, we processed the WebVision dataset similarly to ImageNet-21k-P to remove classes with insufficient images. The resulting dataset, which we denote as WebVision-P, contains approximately 13.5 million training images across 4,189 categories.

### B.3   INATURALIST

iNaturalist Van Horn et al. (2018) contains 2.7 million photographs of organisms in their natural environments, representing 10000 species. The dataset features highly specialized fine-grained categories and natural backgrounds, offering insight into how domain-specific visual features influence alignment scaling.

### B.4   PLACES365

Places365 Zhou et al. (2017) is a large-scale scene-centric dataset containing approximately 1.8 million training images across 365 scene categories. Unlike object-centric datasets such as ImageNet, Places365 focuses on the recognition of environmental scenes, including natural landscapes, urban settings, and indoor environments. Each category includes a wide variety of images to capture the diversity within scene types.

### B.5 INFIMNIST

The MNIST dataset (LeCun et al., 1998) is a classic benchmark in machine learning, comprising 70,000 grayscale images of handwritten digits (0-9), each sized 28×28 pixels. To expand this dataset for more extensive experimentation, we utilize the Infinite MNIST (Infimnist) tool Loosli et al., 2007, which generates additional MNIST-like samples through data augmentation techniques. We create an extended dataset by modifying the original training dataset 19 additional times, resulting in a total of 1.2 million images. This enlarged dataset allows for a more thorough evaluation of scaling effects on the alignment.

## C VALIDATION ON PRIVATE DATA

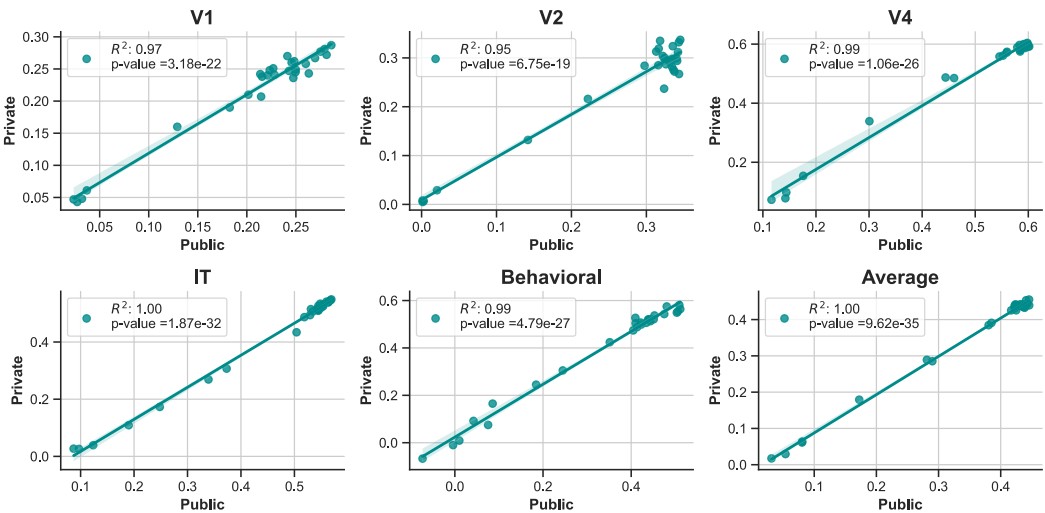

Figure 8: Public benchmarks used in this study correlates highly with private benchmarks on Brain-Score

As described in Section 3 we test a diverse set of models on private benchmarks on Brain-Score platform. All $R^2$ values are above $0.95$ with p-values less than $10^{-18}$.

## D    PRETRAINED MODELS

As part of our comprehensive evaluation, we benchmarked a diverse set of pretrained models sourced from both `torchvision` (maintainers & contributors, 2016) and the `timm` (Wightman, 2019) libraries. We tested a total of 94, including ViT Dosovitskiy et al. (2021), DaViT Ding et al. (2022), LeViT Graham et al. (2021), ConvNeXt Liu et al. (2022a), MobileViT Mehta & Rastegari (2022), MaxVit Tu et al. (2022), FastViT Vasu et al. (2023). Each model varies in parameter count, training sample size, dataset source, and training objective, providing a broad spectrum for analysis.

To verify the generalizability of our findings, we conducted evaluations with these pretrained models, including larger networks like CLIP (Radford et al., 2021) and DINOv2 (Oquab et al., 2023), which are pretrained on richer and more diverse datasets such as LAION Schuhmann et al. (2021; 2022). We also compared variations of these models by examining base pretrained models alongside their fine-tuned counterparts on ImageNet, aiming to investigate the impact of fine-tuning on scaling behavior.

Our results indicate that models with extensive pretraining achieve enhanced behavioral alignment, likely due to their exposure to richer and more varied data. However, similar to models trained solely on ImageNet or EcoSet, these pretrained models still exhibit a saturation effect in neural alignment with the primate visual ventral stream (VVS). This suggests that while larger and more diverse datasets improve behavioral predictability, they do not substantially extend the scaling of neural alignment beyond the observed plateau.

The curves in Figure 9 closely follow the scaling patterns estimated for our trained models shown in Figure 2.c, further validating that the observed saturation is consistent across different pretraining regimes and dataset scales. This reinforces our conclusion that scaling alone is insufficient to overcome the limitations in neural alignment and highlights the need for alternative approaches to improve alignment with neural representations.

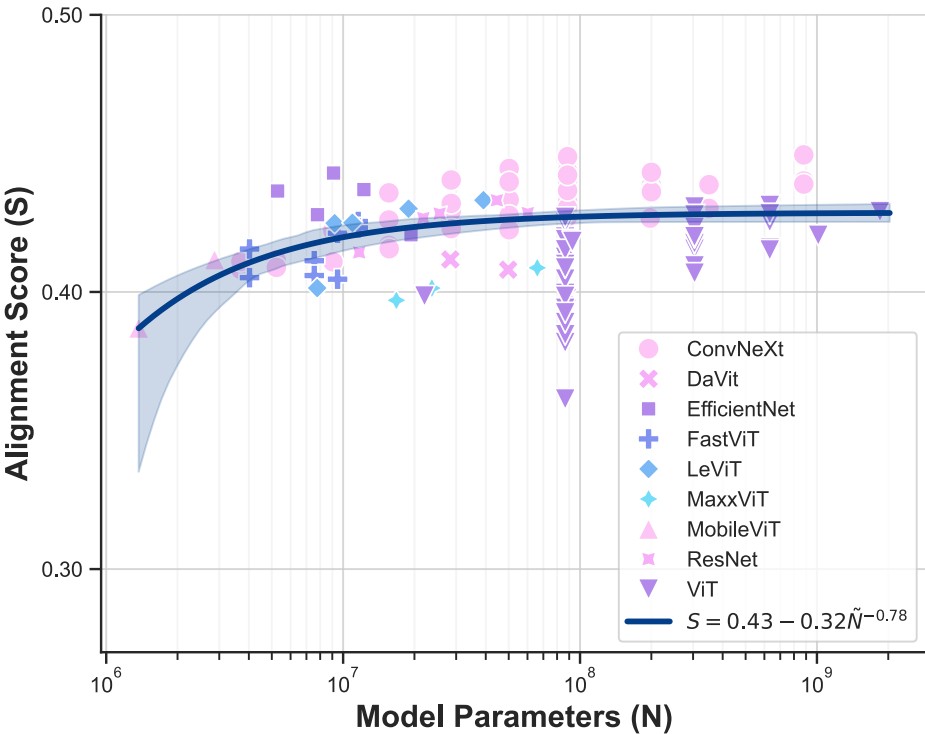

Figure 9: Alignment of pretrained vision models as a function of model parameters

# E    TRAINING EVOLUTION

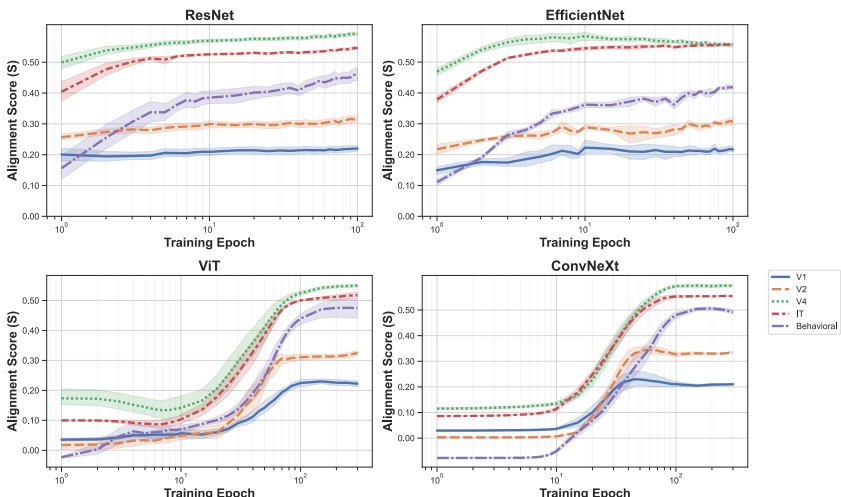

Figure 10: Evolution of per-region alignment throughout training. Models with stronger priors—such as ResNet and EfficientNet—exhibit higher neural alignment initially. However, the gap in representational power diminishes as more generalist models are trained on data.

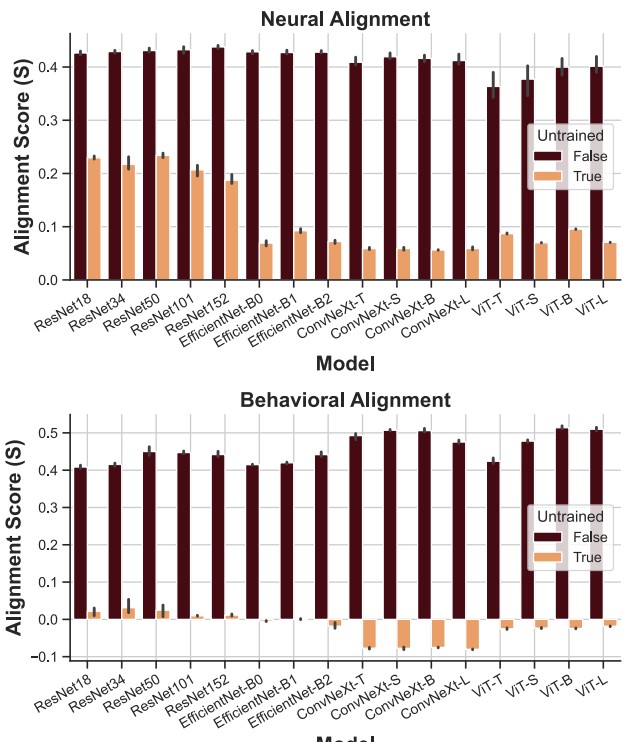

Figure 11: Some untrained models demonstrate non-zero neural alignment at initialization. Nevertheless, all models start with almost zero behavioral alignment, indicating that initial neural alignment arises from architectural biases rather than learned behavior.

## F  EFFECT OF TRAINING OBJECTIVE

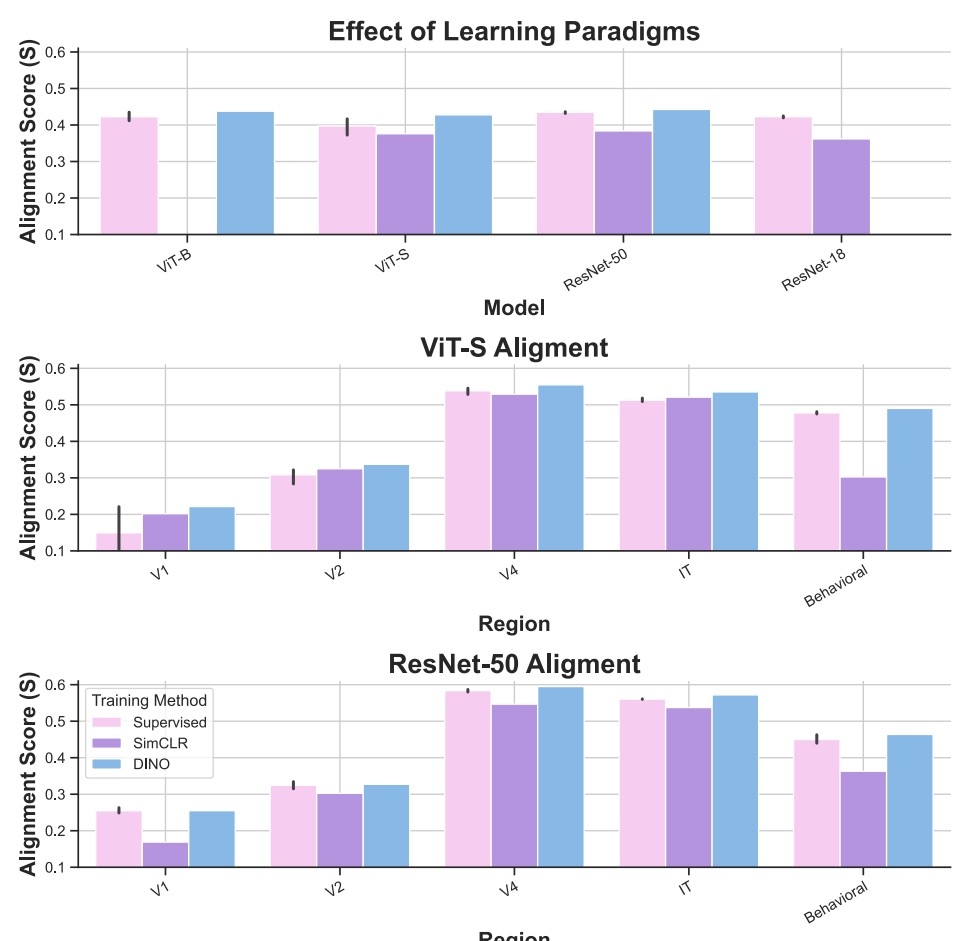

Figure 12: Effect of training objective on alignment

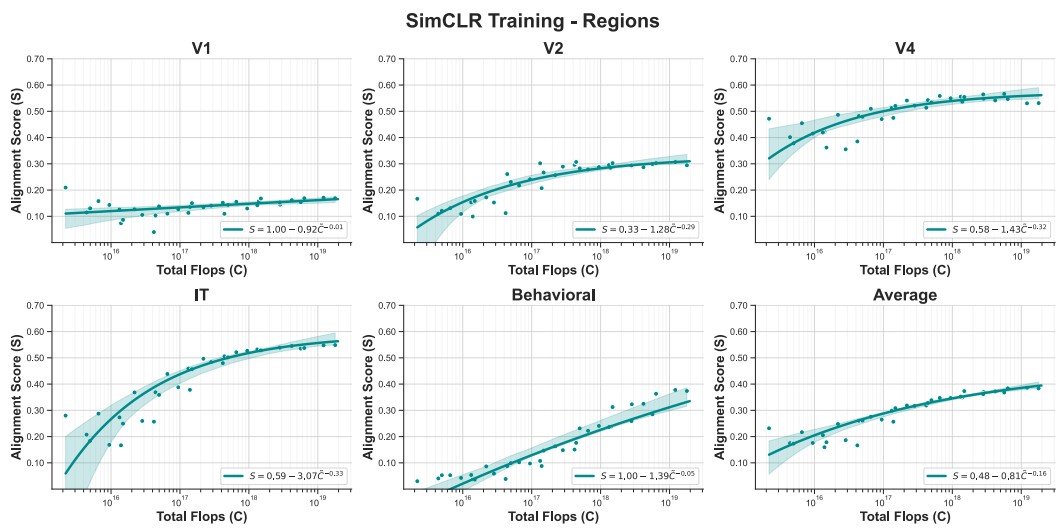

Figure 13: Scaling of SimCLR training across regions

## G    REGION-WISE SCATTER PLOTS

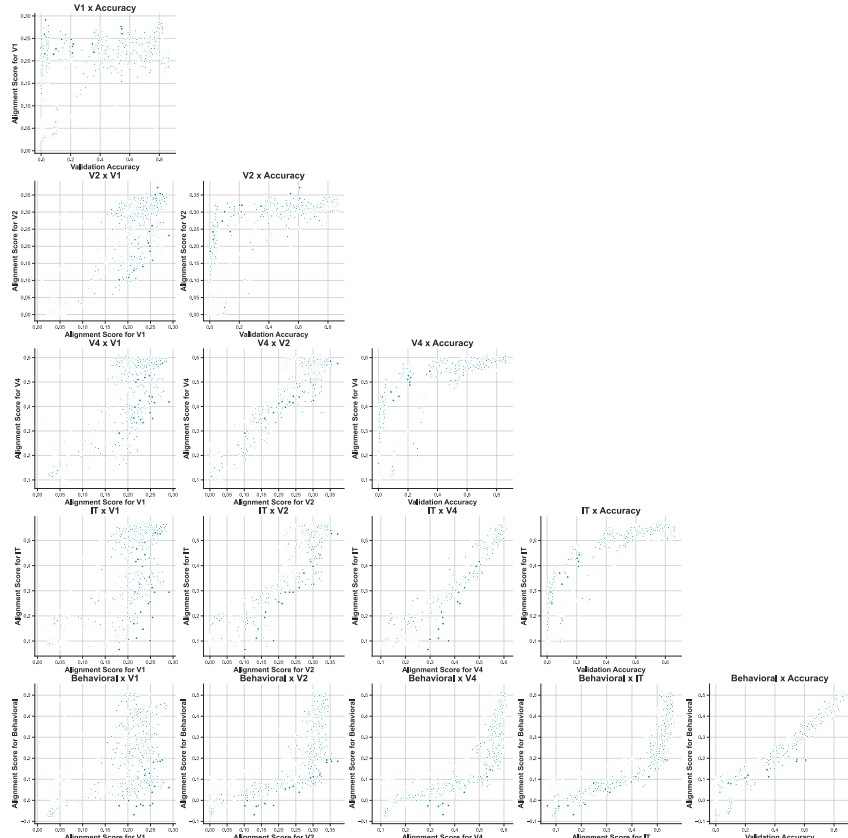

Figure 14: Region vs. Region Comparisons: This figure shows how the alignment scores for each brain region correlate with those of other regions. The diagonal plots illustrate the relationship between the alignment score of each region and the validation accuracy on ImageNet and EcoSet datasets.

