# OpenReview forum: "Scaling Laws for Task-Optimized Models of the Primate Visual Ventral Stream"
_ICLR.cc/2025/Conference — Submitted to ICLR 2025_

### Official Review · Reviewer_rNYX · 2024-10-27

**Soundness:** 3
**Presentation:** 3
**Contribution:** 2
**Rating:** 6
**Confidence:** 3

**Summary:**

This paper explores how varying model sizes impact neural and behavioral alignment, seeking insights into the relationship between model architecture and its ability to mimic human-like neural responses and behaviors.

**Strengths:**

The core claim—model size influencing alignment—is well supported by the results.

Investigating neural and behavioral alignment is a relevant area with potential applications for improving model interpretability and guiding architecture design.

The study contributes to understanding the role of model scale in alignment, a valuable area for both theoretical insights and practical applications in AI research.

**Weaknesses:**

Inductive biases might need better control, either quantitatively or qualitatively, to improve result clarity.

Minor issues: typo at l100 (“ecology”), unclear reference in l130 (“Utah”), and Fig 1 could specify the saturation value.

Benchmark sample size for V1 and V2 is relatively small (315), which may impact result generalizability.

Equation 7’s clarity is limited without referencing equations 8 and 9; introducing C(N, D) = 6ND earlier could help.

**Questions:**

Could there be additional context on the novelty of this work relative to existing literature on model size effects?

Is it possible to control inductive biases more rigorously, either quantitatively or qualitatively?

In Figure 1, what value does alignment saturation reach?

Is “Utah” in l130 a reference or typo?

Would increasing the benchmark sample size for V1, V2 make the results more robust?

Could the paper benefit from additional discussion on neural versus behavioral alignment, and how better control of inductive biases might enhance interpretability?

---

> ### Author Response · Authors · 2024-11-27
> **Response to questions 1-2**
>
> Thank you for your comprehensive and thoughtful review of our manuscript. Below, we address each of your questions in detail.
>
> **1\. Could there be additional context on the novelty of this work relative to existing literature on model size effects?**
>
> Thank you for emphasizing the importance of contextualizing our work within the existing literature on model size effects. Previous work primarily showed scaling with respect to ground-truth performance. Our work evaluates scaling behavior of models with respect to their internal similarity to brain responses, which has to date not been attempted as far as we are aware. Our study introduces several novel contributions that set it apart from previous research:
>
> 1. **Comprehensive Model and Dataset Exploration:** While previous studies have examined specific aspects of model scaling, our work systematically explores over 600 models across various architectures and dataset sizes. This extensive evaluation provides a more holistic understanding of how scaling dimensions interact to influence neural and behavioral alignment.
> 2. **Differential Scaling Laws for Neural and Behavioral Alignment:** Our research identifies distinct scaling laws for neural and behavioral alignment, revealing that while behavioral alignment continues to improve with scale, neural alignment reaches a saturation point. This differentiation offers deeper insights into the limitations of current scaling strategies and their differential impacts on various alignment metrics.
> 3. **Scaling Recipe for Optimal Compute Allocation:** We introduce a novel scaling recipe that optimally allocates compute between model size and dataset size to maximize alignment (Fig 4). This practical guideline is a significant advancement, offering actionable recommendations for future model training strategies aimed at enhancing brain alignment.
> 4. **Granular Analysis Across VVS Hierarchy:** Our study delves into how scaling impacts different regions within the primate visual ventral stream (VVS), from V1 to IT and behavioral outputs (Fig 5). This hierarchical analysis reveals that higher-level regions benefit more from scaling, a detail that had not been thoroughly examined in prior work.
> 5. **Public Release of Extensive Resources:** By open-sourcing our training code, evaluation pipeline, and a vast collection of model checkpoints, we provide invaluable resources for the research community. This transparency facilitates reproducibility and enables other researchers to build upon our findings, thereby accelerating progress in the field.
>
> These contributions collectively advance the understanding of how model scaling influences alignment with both neural and behavioral aspects of the primate visual system, offering new perspectives and practical tools that were not previously available.
>
> **2\. Is it possible to control inductive biases more rigorously, either quantitatively or qualitatively?**
>
> We have expanded the investigation of inductive biases and their impact on alignment in the updated manuscript. Specifically, we analyze the evolution of neural and behavioral alignment during supervised training across different architectures (Figure 8c). Our results confirm that models with strong priors, such as convolutional architectures, exhibit higher neural alignment at initialization compared to more generalist models, like vision transformers.
>
> Additionally, we present a detailed comparison of alignment at initialization across architectures in Figure 11 (Appendix), further supporting the role of inductive biases in early alignment.
>
> We also explore how inductive biases interact with different training objectives in Figure 8d. For example, while the alignment of a ResNet model shows only slight variation between supervised and self-supervised objectives, the alignment of a ViT model is significantly influenced by the training objective. Notably, the self-supervised objective provides a richer learning signal, resulting in a faster rise in alignment during training. This suggests that inductive biases, combined with learning objectives, play a critical role in shaping alignment dynamics.

---

> > ### Author Response · Authors · 2024-11-27
> > **Response to questions 3-5**
> >
> > **3\. In Figure 1, what value does alignment saturation reach?**
> >
> > Thank you for seeking clarification on the alignment saturation values depicted in Figure 1\. In our study, each scaling curve asymptotically approaches a constant value as the scale (compute, model size, or dataset size) increases indefinitely. Specifically:
> >
> > * **Behavioral Alignment:** The alignment score for behavioral alignment approaches a saturation value of **1** in the limit of infinite scaling. This indicates perfect alignment between the model's behavioral predictions and primate behavioral data when unlimited compute and data resources are available.
> > * **Neural Alignment:** The alignment score for neural alignment reaches a saturation value of approximately **0.48**. This plateau suggests that beyond a certain scale, increasing compute, model size, or dataset size yields diminishing returns in terms of improving neural alignment with the primate ventral visual stream.
> >
> > These saturation values are derived from the fitted power-law curves and represent the theoretical maximum alignment achievable under our current model architectures and training datasets.
> >
> > **4\. Would increasing the benchmark sample size for V1, V2 make the results more robust?**
> >
> > We have conducted additional evaluations using more extensive benchmarks available on the Brain-Score platform to assess the robustness of our findings. Figure 8 in the Appendix demonstrates that the results from private benchmarks correlate highly with the public benchmarks used in this study, providing strong evidence of consistency. These supplementary tests validate the reliability of our original findings and suggest that the trends observed in neural and behavioral alignment are robust even when larger sample sizes or additional data are incorporated.
> >
> > **5\. Could the paper benefit from additional discussion on neural versus behavioral alignment, and how better control of inductive biases might enhance interpretability?**
> >
> > In the new "Generalization Beyond Supervised Training" section of the Discussion, we investigate how different training signals influence alignment with the brain and behavior. Figure 7a confirms our findings by showing that in supervised training, neural alignment saturates while behavioral alignment continues to improve with increased compute, with a detailed breakdown presented in Figure 13\. Additionally, our experiments demonstrate that models trained with self-supervised learning methods, such as SimCLR and DINO, achieve similar levels of alignment more efficiently than those trained with supervised learning, particularly for architectures like Vision Transformers that have weaker inductive biases (Figures 7d and 12). This suggests that rich and diverse learning signals facilitate faster and more effective alignment with neural representations. Furthermore, adversarial fine-tuning enhances neural alignment beyond the saturation levels observed with standard training methods (Figure 7b), indicating that introducing adversarial perturbations encourages models to learn more robust features aligned with neural processing.

---

> > > ### Author Response · Authors · 2024-11-27
> > > **Response to weaknesses**
> > >
> > > > Minor issues: typo at l100 (“ecology”), unclear reference in l130 (“Utah”), and Fig 1 could specify the saturation value.
> > >
> > > Thank you for pointing these out, but we believe both are correct: "ecologically viable tasks" refers to tasks that primates would encounter in their natural environment; "Utah array" is the name of the recording device used to retrieve brain data (https://blackrockneurotech.com/products/utah-array/)
> > >
> > > > Equation 7’s clarity is limited without referencing equations 8 and 9; introducing C(N, D) = 6ND earlier could help.
> > >
> > > Thank you for highlighting the clarity issue with Equation 7. We appreciate your suggestion to introduce the relationship C(N,D)=6ND earlier in the manuscript. In the revised version, we have now introduced C(N,D)=6ND prior to describing Equation 7 (lines 210-212).

---

### Official Review · Reviewer_fpoP · 2024-11-04

**Soundness:** 3
**Presentation:** 3
**Contribution:** 2
**Rating:** 6
**Confidence:** 4

**Summary:**

In this paper, the authors study the relationship between the size / compute requirement of popular neural network architectures and their training dataset sizes vs alignment to the biological ventral visual stream. The authors analyze the alignment of various architectures to the primate VVS using the publicly available Brain-Score benchmark and claim that (1) scaling models by increasing parameter count produces diminishing neural alignment beyond a saturation point in model size, but behavioral alignment continues to increase with model size, (2) Alignment scales with training dataset size, (3) Higher visual areas in the cortical hierarchy show stronger gains in alignment with respect to scaling.

**Strengths:**

* This paper sheds light on the similarity of neural network representations to biological visual representations as a function of model size, compute, and training dataset size. The authors have presented these results in a sound theoretical framework by drawing inspiration from analyses of neural scaling laws.
* It is super interesting that different areas of the ventral visual stream have varied effects to scaling of neural architectures/datasets. I have not seen this in prior work to the best of my knowledge and this will raise interesting discussions at ICLR.
* I appreciate that the paper is well-written, the figures are legible and accompanied with reasonably detailed captions.

**Weaknesses:**

* **Lacking evaluation of what model behaviors give rise to alignment.** My main point of feedback to further improve this paper is to address what other factors of artificial neural networks contribute to enhancing similarity to biological vision. It is interesting that there exist scaling laws between model / dataset sizes and neural / behavioral alignment, but this has already been documented in prior studies. I urge the authors to further study the qualitative factors (for e.g. sensitivity to the same spatial frequencies that humans are sensitive to) that give rise to enhanced similarity between ANNs and human vision.
* **Missing evaluation of more recent multimodal models.** There has been a surge in multimodal vision language models that, if evaluated in the same framework established by this paper, would produce really intriguing findings on model scaling and alignment. I encourage the authors to include publicly available large vision language models to increase the impact of their findings, as these VLMs are more widely in use now.

**Questions:**

* Would the authors like to highlight how different training signals would influence alignment to brain / behavior? Humans have a rich multimodal perception of the world, they use depth perception, and predominantly learn without supervision. Are the authors able to tease apart the effects of any such factors in their analyses?

---

> ### Author Response · Authors · 2024-11-27
> **Response to weaknesses**
>
> 1. **Lacking evaluation of what model behaviors give rise to alignment:**
>
> We agree that understanding the specific model behaviors that enhance alignment is crucial for advancing the field. While our current study focused on quantifying scaling laws, we are actively investigating the qualitative factors that contribute to neural and behavioral alignment. One aspect we are exploring is the sensitivity of models to spatial frequencies and other visual features that are relevant to human perception. Additionally, we are analyzing the eigenspectrum of response characteristics of these models to identify patterns that correlate with improved alignment.
>
> 2. **Evaluation of Recent Multimodal Models:**
>
> We acknowledge the growing importance of multimodal vision-language models in current AI research. We have conducted additional evaluations on publicly available large vision-language models, such as CLIP and DINOv2. These models include much larger architectures and are trained on extensive datasets, including LAION, offering a broader scope of training data and objectives compared to our controlled experiments.  Our findings reveal that while these multimodal models achieve enhanced behavioral alignment—likely due to their diverse training objectives and data sources—their neural alignment still exhibits a saturation effect similar to that observed in unimodal models. This indicates that, despite the advanced training paradigms and data diversity, multimodal models do not fundamentally address the limitations in neural alignment scaling.  We have incorporated these results into the revised appendix, including a detailed visualization of the scaling behavior of pretrained models (Figure 9). When compared to Figure 2c, which illustrates the scaling behavior of models we trained, the similar saturation levels observed between pretrained and trained models reinforce the generalizability of our findings. These results provide additional evidence that while scaling improves behavioral alignment, neural alignment requires alternative approaches to overcome the observed limitations.

---

> > ### Author Response · Authors · 2024-11-27
> > **Response to questions**
> >
> > > Would the authors like to highlight how different training signals would influence alignment to brain / behavior? Humans have a rich multimodal perception of the world, they use depth perception, and predominantly learn without supervision. Are the authors able to tease apart the effects of any such factors in their analyses?
> >
> > In the updated manuscript, we further investigate the impact of self-supervised learning methods and adversarial fine-tuning on alignment. Our findings are further corroborated in Figures 7a and 13, where supervised training results align closely with those from self-supervised SimCLR training. As shown in the new Figure 7d, models trained with self-supervised objectives like SimCLR and DINO exhibit different alignment dynamics compared to those trained with supervised learning. Specifically, Vision Transformer models (ViT-S) trained with self-supervised methods achieve similar levels of alignment more efficiently than when trained with supervised objectives. This suggests that the richness and diversity of feedback provided by self-supervised learning facilitate faster and more effective alignment with neural representations. Figure 12 contrasts per region alignment of different objectives, which suggests that certain self-supervised models such as DINO can outperform supervised models in behavioral alignment.
> >
> > Additionally, we examine the effects of adversarial fine-tuning on alignment performance. Our findings indicate that adversarial training can enhance neural alignment beyond the saturation levels observed with standard training methods (Figure 7b). This suggests that introducing adversarial perturbations during training encourages models to learn more robust and generalized features that align more closely with neural representations in the primate visual system.

---

> > > ### Comment · Reviewer_fpoP · 2024-12-03
> > > **Response to rebuttal from reviewers**
> > >
> > > I thank the authors for responding to our reviews. I still retain that the current submission has interesting points for discussion relevant to ICLR and recommend acceptance. I do acknowledge the points of concern from the more critical reviews of this paper, particularly connection to prior works establishing the relationship between architecture / dataset scaling and fit to neural data. I am not changing my score after reading the other reviews and author response, and believe that the paper meets the bar for presentation at ICLR.

---

### Official Review · Reviewer_gmHr · 2024-11-04

**Soundness:** 3
**Presentation:** 4
**Contribution:** 2
**Rating:** 5
**Confidence:** 4

**Summary:**

The paper introduces  a way of calculating scaling laws for neural and behavioral alignment with respect of training data and parameter size of models. It offers an interesting overview of the current status  of models and its performance on these alignment challenges.

**Strengths:**

The paper is well written. The introduction offers a good view of the literature and it is easy to follow the procedure they use to make the evaluation. The results are clearly presented and explained.  It provides a good overview of the current landscape of models in the context of neural and behavioral alignment.

**Weaknesses:**

My main observation about this work is that, while it provides valuable insights and a well-illustrated overview of the current landscape of models and their alignment with  neural of behavioral benchmarks, it could benefit from more clarity on how these findings might guide future advancements. The paper mentions previous work with similar findings, as noted in the discussion; however, it would be helpful to understand more concretely how this work can serve as a foundation for the next steps in the field and how scaling laws can truly help scientists develop the next generation of more brain-like models. For instance what kind of hypothesis can be drawn from scaling laws that can be tested by adding or removing samples/compute of models being constructed to be more brain-like?

Although the limitations section mentions that ‘these functions may not generalize beyond the scales tested,’ this suggests a natural boundary for the impact of these results. Could the authors estimate, based on their scaling laws, what order of magnitude increase in dataset or parameter size might be needed to significantly improve neural alignment beyond the observed plateau?

While I understand that this point is mentioned in the limitations section, I feel it is a significant oversight not to include recurrent models. It is encouraging that the paper mentions that inductive bias in the form of convolution seems to yield faster returns, but this feels limited, given that most of the models tested in these benchmarks are much deeper than what might be expected for an architecture resembling the visual cortex. For instance, would be interesting to see how the scaling laws would apply to CorNet? Is it the case that the more brain like the easier it is to scape the scaling laws? that would be very impactful for the community.


I may have missed it, but did not see mention on self supervised models or robust models and how the scaling laws operate on models trained on these type of frameworks?

**Questions:**

* What are the implications of this work, given the limitations already presented in the paper?

* What would be the predictions for a model that closely resembles the  visual cortex  such as  CorNET ?

* Given that the paper focuses on scaling, Have the authors considered how their scaling laws might apply to or change for models pre-trained on much larger datasets like LAION before fine-tuning on ImageNet? This could provide insights into whether the observed plateaus persist across different pre-training regimes

---

> ### Author Response · Authors · 2024-11-27
> **Response to questions 1**
>
> Thank you for your insightful and constructive review of our manuscript. We are pleased that you found our work well-written and that you recognize the novelty in demonstrating varied scaling effects across different cortical areas. Below, we address each of your comments and questions in detail.
>
> **1\. What are the implications of this work, given the limitations already presented in the paper?**
>
> Given the limitations presented in the paper—such as the specific range of model sizes and dataset volumes examined, the subset of architectures evaluated, and the datasets used—the key implications are as follows:
>
> 1. **Scaling Alone Is Insufficient for Neural Alignment**: The study reveals that while scaling up model parameters and training data consistently enhances behavioral alignment with human performance, it leads to saturation in neural alignment with the primate visual ventral stream (Figs 1b, 5, 7a). This indicates that simply increasing scale using current architectures and datasets is not enough to achieve better neural alignment (Fig 9). The implication is that alternative approaches are necessary to develop models that more accurately mimic neural representations in the brain.
> 2. **Need for Alternative Modeling Approaches**: The observed saturation in neural alignment suggests that future research should explore new strategies beyond traditional scaling (Fig 2, 7a). This includes integrating biologically inspired architectural features such as feedback mechanisms, leveraging additional data modalities, and developing novel training objectives tailored to better capture the dynamics of neural processing.
> 3. **Importance of Inductive Biases**: The findings highlight the significant role of architectural inductive biases in achieving neural alignment. Models with strong inductive biases, like fully convolutional networks (e.g., ResNets and EfficientNets), demonstrate higher initial neural alignment even before training (Figs 2, 7c, 10, 11). This implies that incorporating architectural priors that reflect biological neural structures can improve alignment efficiency without solely relying on scaling.
> 4. **Guidance for Resource Allocation**: By introducing and fitting parametric power-law scaling laws, the study provides a predictive framework for how alignment scales with compute and data (Fig 4a). This quantitative approach offers practical guidance on how to allocate computational resources effectively between model complexity and dataset size to optimize both neural and behavioral alignment.
> 5. **Potential of Adversarial Training and Alternative Learning Signals**: The experiments with adversarial fine-tuning show its potential in enhancing neural alignment beyond the saturation levels observed with standard training methods (Fig 7b). This suggests that incorporating robust training approaches and alternative learning signals could play a crucial role in developing models that better align with neural data.
> 6. **Differential Impact Across Brain Regions**: The discovery of an ordered effect of scaling on alignment across different brain regions provides deeper insights into how scaling differentially affects various levels of neural processing (Fig 5). This suggests that scaling strategies may need to be tailored to target specific regions within the visual cortex to achieve optimal alignment.
>
> In summary, the implications of this work emphasize that while scaling is beneficial for improving behavioral alignment, it is not sufficient for advancing neural alignment with the brain's visual system using current models and datasets. This underscores the necessity of exploring new modeling approaches that incorporate biological principles and alternative training strategies. Despite the limitations, such as the specific models and datasets used, these findings offer valuable insights and directions for future research aimed at bridging the gap between artificial neural networks and biological neural processing.

---

> > ### Author Response · Authors · 2024-11-27
> > **Response to questions 2-3**
> >
> > **2\. What would be the predictions for a model that closely resembles the visual cortex such as CorNET?**
> >
> > Thank you for your suggestion to consider recurrent and more biologically inspired models like CORNet. In our experiments, we included CORNet-S, which appears in several figures’ legends (Figs 2c, 3b, 4a-b, 6a) except where a specific model architecture was highlighted. Our results show that CORNet-S exhibits scaling characteristics similar to ResNet models. Specifically, CORNet-S follows the same trend: neural alignment plateaus as scale increases, while behavioral alignment continues to improve.  This finding suggests that recurrence alone, as implemented in CORNet-S, does not inherently address the scaling limitations for neural alignment. To achieve better neural alignment, it may be necessary to incorporate additional biological principles or mechanisms beyond those currently represented in standard ventral stream-inspired architectures.  As a future direction, we propose investigating models with stronger biological constraints, such as VOnenet, which integrates biologically plausible features like in V1 of primate VVS.
> >
> > **3\. Given that the paper focuses on scaling, Have the authors considered how their scaling laws might apply to or change for models pre-trained on much larger datasets like LAION before fine-tuning on ImageNet? This could provide insights into whether the observed plateaus persist across different pre-training regimes**
> >
> > We have conducted evaluations with 94 pre-trained models from timm library to verify the generalizability of our findings (Fig. 9). These neural networks include CLIP and DINOv2 models, which are larger than our largest trained models and are pre-trained on richer, more diverse datasets such as LAION. We also compared variations of these models, such as a base pre-trained model and its fine-tuned counterpart on ImageNet, to investigate the impact of fine-tuning on scaling behavior.  Our results show that models with extensive pretraining achieve enhanced behavioral alignment, likely due to their exposure to richer and more varied data. However, similar to the models trained solely on ImageNet or EcoSet, these pre-trained models still exhibit a saturation effect in neural alignment with the primate visual ventral stream (VVS). This indicates that while larger and more diverse datasets improve behavioral predictability, they do not substantially extend the scaling of neural alignment beyond the observed plateau.  In the revised appendix, we provide detailed visualizations of the scaling behavior of pre-trained models in Figure 9\. These curves closely follow the scaling patterns estimated for our trained models in Figure 2c, further validating that the observed saturation is consistent across different pre-training regimes and dataset scales. This reinforces our conclusion that scaling alone is insufficient to overcome the neural alignment limitations and highlights the need for alternative approaches.

---

> > > ### Author Response · Authors · 2024-11-27
> > > **Response to weaknesses**
> > >
> > > > I may have missed it, but did not see mention on self supervised models or robust models and how the scaling laws operate on models trained on these type of frameworks?
> > >
> > > #### **1\. Self-Supervised Models**
> > >
> > > To investigate SSL scaling, we conducted additional experiments using SimCLR (Contrastive Learning) across various model and data scales. In the revised manuscript, **Figure 7a** illustrates the scaling curves for SimCLR models trained on subsets of ImageNet. Additionally, **Appendix Figure 13** provides a per-region breakdown of alignment during SimCLR training.
> > >
> > > **Findings:**
> > >
> > > * **Neural Alignment:** Self-supervised models exhibit similar saturation in neural alignment as supervised models. Although SSL enhances the richness and diversity of learned representations, it does not fundamentally alter the scaling laws governing neural alignment with the primate visual ventral stream.
> > > * **Behavioral Alignment:** Consistent with supervised models, self-supervised models show continuous improvements in behavioral alignment with scale, following a power-law relationship without noticeable saturation within the tested ranges.
> > >
> > > **Implications:**
> > > These results demonstrate that SimCLR models provide comparable improvements in behavioral alignment to those achieved by supervised learning. However, the persistent saturation in neural alignment suggests that scaling alone, regardless of the learning paradigm, is insufficient to achieve higher neural alignment with the primate visual system.
> > >
> > > #### **2\. Robust Models**
> > >
> > > We also explored the impact of adversarial training, a robust learning approach, on scaling behavior. Our preliminary experiments focused on adversarial fine-tuning of existing models, as opposed to training adversarially from scratch. The revised manuscript now includes **Figure 7b**, which illustrates the scaling curves of adversarially fine-tuned models.
> > >
> > > **Findings:**
> > >
> > > * Adversarial fine-tuning improved both neural and behavioral alignment along the scaling curves estimated for non-adversarially trained models.
> > >   * These improvements suggest that adversarial training can raise the saturation levels of neural alignment more effectively than conventional training.
> > >   * Our initial results also indicate that adversarial fine-tuning is more compute-efficient compared to adversarial training from scratch.
> > >
> > > **Implications:**
> > > These findings highlight adversarial training as a promising direction for overcoming the limitations of current scaling laws in neural alignment. While further investigation is needed, the initial evidence suggests that robust learning techniques may push higher or help break through the observed saturation levels.

---

> > > ### Comment · Reviewer_Z99m · 2024-12-02
> > > **Response.**
> > >
> > > This is great, thanks.

---

> > ### Comment · Reviewer_Z99m · 2024-12-02
> > **Response**
> >
> > Thanks for this in depth response.
> >
> > I think that the following points are all closely related: **Scaling Alone Is Insufficient for Neural Alignment**, ****Need for Alternative Modeling Approaches**, **Importance of Inductive Biases**. These all are aimed at bringing the stalled progress of NeuroAI into greater focus. I believe this is a great point, but it is no longer novel. I think the current manuscript is overly indexed to the plateau of NeuroAI, and not enough attention is paid to how to escape it.
> >
> > **Guidance for Resource Allocation**, **Potential of Adversarial Training and Alternative Learning Signals**: I agree this could be big. But only if scaling laws are a path forward for NeuroAI. As mentioned I'm excited about the result in Fig. 6 about scaling laws for behavior. This could be the sole focus of the paper if scaling laws indicate we need $X to fully reverse engineer behavior (and $X was not some unrealistic number). I'm not sure that's the case, however. Even in the adversarial training case where the authors show a small boost in alignment, the slope looks extremely flat. If we were to scale up to 10^100 flops, the alignment score would reach 0.51, according to the equation. Again, this is the null result bit that I don't think is novel. We need new ways forward.
> >
> > **Differential Impact Across Brain Regions** Thanks for this response.

---

> > ### Comment · Reviewer_gmHr · 2024-12-03
> >
> > I want to thank the authors for such in depth response.  I may have to say that I align with reviewer Z99m, the point on "Scaling Alone Is Insufficient for Neural Alignment:" has been raised before and unfortunately is not a novel contribution. I was very optimistic by the point raised in "3. Importance of Inductive Biases" but I feel the evidence is rather limited. According to the results on CorNet, it is not super clear that a more "biologically plausible" model can scape the scaling laws easily, and there is little understanding on how to assess these inductive bias, through the lens of the scaling laws, such that is clear the way forward.
> >
> > Thank you for providing more results on point 5 and 6. I think there could be something interesting if the scaling laws can be calculated for robust models, or at least an understanding that can motivate novel experiments and efforts on the model and data side to move beyond the saturation mark.

---

### Official Review · Reviewer_Z99m · 2024-11-04

**Soundness:** 2
**Presentation:** 4
**Contribution:** 2
**Rating:** 5
**Confidence:** 4

**Summary:**

The authors investigate so-called neural scaling laws for predicting visual behavior and neural activity. "Scaling laws" are empirical trends that show a relationship between model scale (e.g., compute used or amount of data used in training) and its loss on a pretraining task. Here, the authors show different functional forms of scaling laws for predicting neural activity vs. behavior, where the latter is far more promising than the former.

**Update**
I'm on the fence with this paper. I think there's tons of well-done experiments, and I think the message is important to the field of NeuroAI albeit not totally a novel one. I think the line fits are also still problematic and telling a story that's not totally backed up by the data, although I appreciate that the authors are trying to establish a parallel with work in AI on scaling laws. If there were a clear direction forward then this would be a no-brainer accept. As is, I believe it's borderline. I am increasing my score to reflect this.

Also on a separate note, my apologies to the authors for neglecting to respond to all of their points. I was confused by the threading of the responses and mistook the authors' responses to gmHr for responses to my own questions.

**Strengths:**

The authors completed an extensive sweep through model architectures, compute, and data budgets, in order to give a detailed view of how model scale relates to neural and behavioral brain scores. The key findings here are important (although with debatable novelty): (1) Neural fits asymptote or worsen with scale, (2) behavioral fits are linear with scale (although scale alone appears to be insufficient), (3) the ceiling and form of scaling laws is different for each visual area region that was investigated. Overall, this is a nice capstone on BrainScore, and perhaps is most notable for showing how methods from AI are not always applicable for explaining brain and behavior.

**Weaknesses:**

1. The power of scaling laws in domains like language (IMO) is that they imply "all you need is scale." That is, and in the spirit of the bitter lesson, there are no conceptual barriers to achieving a criterion level of performance, only engineering ones. If this were the case in brain science it would be a true world changer. But as this paper (and others which were cited) show, this is not the case. DNNs + scale are not the solution to explaining the variance in brainscore visual system recordings. In that sense I see a large overlap between the findings and result of [1] in which they found a trade-off between ImageNet performance and BrainScore fits. In both cases, these are null results. It is great to show this result, but the lack of a direction forward is concerning.

To drive the point home, in Fig 3, the authors show that training on ImageNet21k (but curiously not WebVision which has more images) leads to better fits. Indeed this would seem to be a scaling law... but the effect size makes it impractical at best: the model maxes out around 0.45 alignment even after all of that data.

For these reasons I struggle to see how this paper makes a strong contribution to the field. It feels better served as a memo or blog post than a conference or journal paper.

2. I think some of the line fits are overly optimistic. For example, in Fig 1, the neuro line is monotonically increasing. But if I squint and just look at the dots, it looks more like a subtle decrease in fits, on average, as a function of compute. This issue is in many of the plots. This relates to my thoughts in (1) about what this all means and whether or not the findings are novel. See fig 2 ViT behavioral line fits for an example where it's not just for neural data. I am marking down the "Soundness" of the paper because of these line fits, but to be honest I don't have any great suggestions about how to improve the fits while maintaining interpretable "laws" when you have what look like non-monotic changes like with the Neural data in Fig 1c.

3. The y limits of the plots should be fixed to one range. It looks like 0-0.7 captures everything. Theres too much bouncing around between different ranges in different subplots. Also could you label what dataset the validation accuracy is derived from on plots where you report it?

[1] Linsley et al. Performance-optimized deep neural networks are evolving into worse models of inferotemporal visual cortex.

**Questions:**

1. On Figure 6b, that's a beautiful correlation. How far can you take it out? Just eyeballing I'd guess it would get near 0.7. Perhaps a pivot for the paper, to get the positive result I think it needs, would be to focus on this scaled-up model of behavior? Just a thought.

2. Why do you think neural scaling laws are different for different brain regions and also for behavior? This is a complex question of course, and I don't expect a definitive answer, but perhaps there's something interesting here.

---

> ### Author Response · Authors · 2024-11-27
> **Response to weaknesses 1**
>
> We sincerely appreciate your review and the time you invested in evaluating our manuscript. We strongly feel that this submission is a significant contribution to the NeuroAI field and that it should thus be featured at ICLR. We address your concerns and questions below.
>
> **1\. Novelty and Contribution to the Field**
>
> *Concern:* You expressed concern about the novelty of our findings and how our work contributes to the field, noting similarities with prior studies such as Linsley et al.
>
> *Response:* Previous studies have indeed explored the relationship between task performance and brain alignment. Almost all of them found a continued positive relationship. We are only aware of two papers (Schrimpf et al. 2018 and Linsley et al.) that raised the point this relationship might break for data from a single visual area (IT). Our work provides substantially novel findings in several key aspects:
>
> * **Dissociation Between Neural and Behavioral Alignment:** Our findings highlight a clear dissociation between neural and behavioral alignment as models scale (Figs. 1b, 5, 7a, 13). While behavioral alignment continues to improve, neural alignment saturates—a phenomenon not quantitatively characterized in prior work.
> * **Unexpected saturation**: Across many domains of brain function, larger and more task-performant models lead to improved alignment with brain data (e.g. vision \[Yamins et al. 2014\], auditory \[Kell et al. 2018\], language \[Schrimpf et al. 2021\], motor \[Vargas et al. 2024\]). It is thus reasonable to believe that continued performance scaling will yield continued brain alignment gains, and in our experience this is the reality for most of the field; for instance virtually all models on Brain-Score are pre-trained machine learning models. We show that scaling the ML way will not improve alignment to the brain’s visual system, and pinpoint the primary failure cases to early visual processing (see below).
> * **Graded Effect Across Brain Regions:** We uncover an ordered effect of scaling on alignment across different brain regions in the visual hierarchy (V1, V2, V4, IT; Figs 5, 13), providing insights into how scaling differentially impacts various levels of neural processing.
> * **Systematic Quantification:** We provide a systematic and controlled investigation into how scaling both model size and dataset size affects neural and behavioral alignment. By training over 600 models under controlled conditions, we eliminate confounding factors present in studies using pre-trained models with varying architectures and training regimes.
> * **Parametric Scaling Laws:** We introduce and fit parametric power-law scaling laws to our data, offering a predictive framework for how alignment scales with compute and data. This quantitative approach allows us to extrapolate and predict alignment at scales beyond those directly tested. Indeed, we further validated these predictions with unsupervised, multimodal, and adversarially trained variants (new Figs 7ab, 9, and 13).
> * **The Role of Inductive Biases**: Our analysis demonstrates how inductive biases in neural network architectures (e.g., convolutions in ResNets vs. transformers) affect alignment. In the revised manuscript, we present training dynamics (Figs. 2, 7c, 10, 11 ) showcasing how models with differing biases converge to similar representations over time, albeit starting from distinct initial points. This sheds light on the influence of architectural priors on neural and behavioral alignment.
> * **Direction Forward:** Our study provides quantitative scaling laws that highlight how computational resources can be effectively allocated between model size and dataset size to optimize neural and behavioral alignment. While these findings underscore the benefits of scaling especially for behavioral alignment (a positive finding), the observed saturation in neural alignment suggests that scaling alone is insufficient to achieve more accurate models of the primate visual ventral stream.
>   We discuss the necessity of exploring alternative strategies in Discussion *Limitations and Future Directions* section, such as integrating biologically inspired architectural features (e.g., V1 block in VOneNet), co-training with brain data, and developing novel training objectives tailored to better capture neural dynamics. Additionally, our experiments with adversarial fine-tuning demonstrate its potential in raising alignment saturation levels, suggesting that robust training approaches could play a crucial role. As such, combining stronger inductive priors with advanced training paradigms like adversarial fine-tuning offers a promising path toward developing next-generation models that more faithfully mimic biological vision systems.
>
> We believe these contributions offer new insights and a valuable direction for future research, emphasizing that scaling alone may not suffice to improve neural alignment, thereby highlighting the need for novel modeling approaches.

---

> > ### Author Response · Authors · 2024-11-27
> > **Response to weaknesses 2**
> >
> > **2\. Line Fits and Interpretation of Results**
> >
> > *Concern:* You noted that some of the line fits, particularly for neural data, may be overly optimistic and may not accurately reflect non-monotonic trends in the data.
> >
> > *Response:* Thank you for bringing this to our attention. We selected power-law curves based on their widespread use and interpretability in machine learning scaling law literature. However, we recognize that alternative parametric forms might better capture the nuances of our data. We are open to exploring other functional forms, such as sigmoid functions or piecewise linear models, to potentially provide a better fit to the observed trends. Nonetheless, our choice was motivated by the balance between fit quality and the ability to derive meaningful scaling exponents, which facilitate the optimization of compute allocation. We agree that the variability in the neural alignment data warrants careful consideration. In response, we have added bootstrapped confidence intervals to our plots to represent the variability and uncertainty in the fits.
> >
> > We believe these contributions offer new insights and a valuable direction for future research, emphasizing that scaling alone may not suffice to improve neural alignment, thereby highlighting the need for novel modeling approaches.

---

> > > ### Author Response · Authors · 2024-11-27
> > > **Response to questions**
> > >
> > > *a) On Figure 6b, could focusing on the scaled-up model of behavior strengthen the paper?*
> > >
> > > *Response:* Thank you for your positive remark about Figure 6b. We are pleased that you find this aspect of our work engaging. We understand that you suggest focusing on the "scaled-up model of behavior" to strengthen the paper. Could you please clarify what you mean by "scaled-up model of behavior"? Are you proposing that we place greater emphasis on the behavioral alignment achieved at larger scales, or perhaps delve deeper into how scaling impacts behavioral predictions of the models?
> > >
> > > We greatly appreciate your insightful feedback and look forward to your clarification to help us enhance our manuscript further.
> > >
> > > *b) Why are neural scaling laws different for different brain regions and behavior?*
> > >
> > > *Response:* The differing scaling laws across brain regions and behavior likely stem from the distinct computational functions and complexities associated with each region. Higher-level areas like IT and behavioral outputs involve more abstract and integrative processing, which may benefit more from increased model capacity and data diversity. In contrast, early visual areas like V1 and V2 process more basic visual features and may reach an alignment plateau as they are already well-modeled by simpler architectures or smaller scales. Additionally, the inductive biases inherent in certain architectures may align more closely with the computational principles of specific brain regions, influencing how scaling affects their alignment. In the revised manuscript, we further investigate how inductive biases of models influence the alignment at initialization and during training (Fig 7c, 10, 11).

---

> > > > ### Comment · Reviewer_Z99m · 2024-12-02
> > > > **Clarification**
> > > >
> > > > **Response: Thank you for your positive remark about Figure 6b. We are pleased that you find this aspect of our work engaging. We understand that you suggest focusing on the "scaled-up model of behavior" to strengthen the paper. Could you please clarify what you mean by "scaled-up model of behavior"? Are you proposing that we place greater emphasis on the behavioral alignment achieved at larger scales, or perhaps delve deeper into how scaling impacts behavioral predictions of the models?**
> > > >
> > > > I want to know if scale is all you need in order to reverse engineer human behavior on the psychophysics task. That would be truly remarkable. By eye though, it looks like as accuracy on those tasks approaches 100% (is this even a reasonable number to reach?) then behavioral alignment maxes out ~0.7. It would be nice to see if that is indeed the case and if any architectural choices/mechanisms can increase/decrease the likelihood of developing a complete model.

---

### Author Response · Authors · 2024-11-27
**General response #1**

We sincerely appreciate the time and effort the reviewers have invested in reviewing our manuscript. Their insightful comments and constructive feedback have been invaluable in improving the quality and clarity of our work.

**Summary of Key Additions to the Updated Manuscript:**

1. **Improving Robustness of Curve Fits via Confidence Intervals:**
   * To address concerns about the variability and reliability of our curve fits, we now include 95% confidence intervals estimated from 1,000 bootstrapped samples for all scaling curves. This statistical enhancement provides greater confidence in our findings and helps verify the robustness of our scaling laws.
   * Additionally, we have reworked all figures to use coherent color palettes and improved readability. These visual refinements ensure that the figures are more accessible and easier to interpret, facilitating a clearer understanding of our results.
2. **Impact of Inductive Biases on Alignment Dynamics:**
   * We have elaborated on how inductive biases in neural network architectures influence alignment. Our analysis now includes additional figures (Figures 7c, 10, 11\) demonstrating that models with strong inductive biases, such as fully convolutional networks like ResNets and EfficientNets, exhibit higher initial neural alignment. This insight sheds light on the importance of architectural choices in achieving efficient and effective alignment with neural data.
3. **Influence of Different Training Signals:**
   * We have investigated how different training signals, including self-supervised learning methods like SimCLR and DINO, and adversarial fine-tuning, impact alignment with the brain and behavior. Our findings, presented in Figures 7a, 7b, 7d, 12, and 13, show that these training strategies can enhance alignment, particularly for models with weaker inductive biases. This suggests that rich and diverse learning signals facilitate faster and more effective alignment with neural representations.
4. **Evaluation of Pretrained and Multimodal Models:**
   * We have extended our analyses to include evaluations of larger pretrained models and multimodal models (e.g., CLIP and DINOv2) trained on extensive datasets like LAION. The results, detailed in Figure 9, indicate that they exhibit saturation in alignment confirming our earlier results in Figure2. This reinforces our conclusion that scaling alone is insufficient to overcome the limitations in neural alignment and highlights the need for alternative approaches.
5. **Additional Discussion on Future Directions:**
   * We have added a concise section outlining potential future research avenues. These include exploring adversarial training to push neural alignment beyond current saturation levels, leveraging biologically inspired architectures to develop more compute-efficient models, and investigating co-training with brain data to enhance alignment with neural representations.
6. **Clarifications and Corrections:**
   * We have addressed specific points raised by the reviewers, such as clarifying the alignment saturation values in Figure 1, discussing the impact of scaling on different brain regions (Figure 5), and correcting typographical errors. We have also provided more context on the novelty of our work relative to existing literature and emphasized the practical implications of our findings.

We believe that these additions and revisions have strengthened our manuscript by providing deeper insights into the mechanisms underlying neural and behavioral alignment, and by addressing the key concerns raised in your reviews. We kindly ask you to consider these new analyses and enhancements when evaluating our work.

Your thoughtful feedback has been instrumental in refining our study, and we are grateful for your contributions to improving the quality of our research. We hope that the improvements we have made not only address your concerns but also demonstrate the significance and novelty of our contributions to the field.

Thank you once again for your time and consideration.

---

### Meta-Review · Area_Chair_J9Bn · 2024-12-22

**Metareview:**

The paper dives deep into the alignment of neural network models and neural response patterns in the visual ventral system. After checking myself, the quality of the paper and the visuals is top, and the findings are indeed intriguing and thought provoking, showing a great deal of work and craftmanship. On the other hand, the majority of reviewers point out that this is definitely positive, but the paper is missing actionable insights that set it apart from other papers, which have already published on this direction. This, with the fact that there were 7 (!!) updates on the manuscript point to having this paper revised and resubmitted so that the paper shines.

**Additional Comments On Reviewer Discussion:**

"I appreciate the efforts from the authors and I think it exhibits a lot of careful thought and diligent work. My main concern is that the core claim of the paper seems to have limited impact in the field, it is not clear how to move forward. The study then turns into assessing the ground in current architectures, which seem have been addressed in other pieces of work previously published. There are few really good leads that can turn this into a very impactful paper, but currently it seems may be limited for this venue. Looking forward to hear from the other reviewers."

"Agree with this take. On the one hand, the novelty is debatable and it doesn't provide a path forward. On the other hand, I can imagine citing this paper and the experiments are well done (though I still have issues with the line fits/scaling laws)."

---

### Decision · Program_Chairs · 2025-01-22

Reject